# Weakly Supervised Nuclei Segmentation with Point-Guided Attention and Self-Supervised Pseudo-Labeling

**DOI:** 10.3390/bioengineering12010085

**Published:** 2025-01-17

**Authors:** Yapeng Mo, Lijiang Chen, Lingfeng Zhang, Qi Zhao

**Affiliations:** Institute of Electronic Information Engineering, Beihang University, 37 Xueyuan Road, Haidian District, Beijing 100191, China; sy2202122@buaa.edu.cn (Y.M.); chenlijiang@buaa.edu.cn (L.C.); zy2302502@buaa.edu.cn (L.Z.)

**Keywords:** weakly supervised learning, nuclei instance segmentation, multi-scale Gaussian kernel, point-guided attention, pseudo-label updating

## Abstract

Due to the labor-intensive manual annotations for nuclei segmentation, point-supervised segmentation based on nuclei coordinate supervision has gained recognition in recent years. Despite great progress, two challenges hinder the performance of weakly supervised nuclei segmentation methods: (1) The stable and effective segmentation of adjacent cell nuclei remains an unresolved challenge. (2) Existing approaches rely solely on initial pseudo-labels generated from point annotations for training, and inaccurate labels may lead the model to assimilate a considerable amount of noise information, thereby diminishing performance. To address these issues, we propose a method based on center-point prediction and pseudo-label updating for precise nuclei segmentation. First, we devise a Gaussian kernel mechanism that employs multi-scale Gaussian masks for multi-branch center-point prediction. The generated center points are utilized by the segmentation module to facilitate the effective separation of adjacent nuclei. Next, we introduce a point-guided attention mechanism that concentrates the segmentation module’s attention around authentic point labels, reducing the noise impact caused by pseudo-labels. Finally, a label updating mechanism based on the exponential moving average (EMA) and k-means clustering is introduced to enhance the quality of pseudo-labels. The experimental results on three public datasets demonstrate that our approach has achieved state-of-the-art performance across multiple metrics. This method can significantly reduce annotation costs and reliance on clinical experts, facilitating large-scale dataset training and promoting the adoption of automated analysis in clinical applications.

## 1. Introduction

Histopathological images play a crucial role in pathological diagnosis [1]. By analyzing these images, pathologists can devise appropriate treatment plans and prognosis evaluations for patients [2,3]. As a computer-aided technique, nuclei segmentation plays a crucial role in pathological image analysis, such as revealing tumor differentiation status [4] and cellular heterogeneity [5] and evaluating the tumor microenvironment [6], including immune cell infiltration [7] and angiogenesis [8]. Additionally, compared to manual methods, it significantly reduces the costs of pathological image analysis, shortens the analysis time, and enhances the efficiency and accuracy of medical image processing. It also aids pathologists in better analyzing nuclear morphology and structure, enabling early disease detection, quantitative assessment, and scientific research [9,10,11].

Traditional nuclei segmentation methods primarily include level set methods [12], watershed algorithms [13,14], edge detection, clustering [15], and thresholding techniques. These methods typically require extensive manual pre-processing and post-processing steps, such as noise removal and morphological operations. Additionally, they have a strong dependency on parameter settings, making them difficult to adapt to complex nuclear shapes and variations, resulting in highly unstable predictions.

In recent years, deep learning methods have been widely applied in medical image processing due to their ability to efficiently handle large-scale data and exhibit strong generalization capabilities. Semantic segmentation networks [16], like U-Net [17], are used to localize and segment nuclei in images, achieving success in medical image segmentation with their straightforward and effective architectures. Micro-Net [18] builds on U-Net by incorporating multi-scale feature extraction to enhance the detection capability for nuclei of varying sizes. The Vision Transformer (ViT) [19] is one of the most significant achievements in recent years for visual tasks. It applies the Transformer model [20], originally designed for natural language processing, to the field of image analysis, challenging the dominance of convolutional neural networks (CNNs). It enables the use of attention mechanisms in image segmentation. Subsequently, numerous ViT-CNN hybrid networks, such as Ds-TransUNet [21] and CellDETR [22], have been applied to the field of cell detection. With the advent of the era of large models, general segmentation models like SAM [23] and OMG-Seg [24] are becoming the next focus for researchers. However, these semantic segmentation methods cannot differentiate between different instances of the same category, such as multiple cell nuclei within one image. In contrast, instance segmentation networks [25,26,27] employ more rigorous labeling to precisely annotate the boundaries of each cell nucleus, distinguishing closely adjacent or overlapping nuclei. However, fully supervised methods for cell nuclei instance segmentation rely heavily on extensive pixel-level annotation data, which are time-consuming to acquire and annotate, often requiring expert guidance. Additionally, these approaches may encounter overfitting issues when confronted with the intricate morphological and structural variations inherent in cell nuclei.

Compared to fully supervised methods for nuclei segmentation, weakly supervised approaches have gained attention due to their simplicity and cost-effectiveness in acquiring data labels. However, these methods sacrifice label accuracy. Effectively leveraging ambiguous annotation information to enhance model performance remains a major challenge in weakly supervised nuclei segmentation. Weakly supervised annotations commonly include box, point, and image-level annotations:Box annotations simplify the annotation process by drawing a rectangular box around each instance. BoxInst [28] replaces the original pixelwise mask loss with projection loss and pairwise affinity loss, achieving promising results using only box annotations. Building on BoxInst, BoxTeacher [29] introduces a mask-aware confidence score to estimate the quality of pseudo-labels, along with noise-aware pixel loss and denoising affinity loss to adaptively optimize the pseudo-labels for the student network. Wang et al. developed a polar-transformation-based MIL strategy to improve segmentation with loose bounding box supervision [30]. Their method emphasizes pixels closer to the polar origin, achieving robust segmentation performance even with imprecise bounding boxes. These methods are suitable for objects with clear boundaries, but it is generally less effective for objects with complex shapes or those that are densely packed. Moreover, this method still requires relatively complex annotation efforts.Image-level annotations are the weakest form of annotation in weakly supervised learning. Unlike other methods that describe specific regions or object details within the image, this approach only uses labels that describe the overall content of the image, for example, classification labels, or even labels distinguishing between positive and negative samples in an image. MICRA-Net [31] leverages latent information from the trained model and uses gradient class activation maps to generate detailed feature maps, reducing the need for expert annotations. Zhou et al. employed cyclic learning [32] based on multi-task learning, alternating between classification and semi-supervised tasks. Their method achieves performance close to fully supervised approaches and is compatible with different backbones and segmentation architectures. While image-level label-based methods are efficient for annotation, they do not provide spatial information or the boundary details of instances, which leads to suboptimal performance in tasks like nuclei segmentation.Point annotations [33,34,35,36,37,38] not only facilitate easy label generation but also provide spatial location information of the nuclei, thus presenting an excellent alternative. Given that point annotations do not provide information about the boundaries of nuclei, existing methodologies leverage Voronoi diagrams and k-means clustering algorithms [39] to generate pseudo-labels (as illustrated in Figure 1). Notably, Qu’s method [40,41] successfully reduced the annotation time on a custom dataset from 114 min to 15 min. Some methods [35,42] employ the Sobel filter to generate pseudo-boundary maps for nuclei boundary refinement. Liu et al. [43] directly processed the segmentation prediction map for more stringent instance segmentation boundary labels, feeding them into an instance segmentation network for learning. SC-Net [44] uses a co-training strategy and self-supervised learning to improve segmentation accuracy, achieving strong results on MoNuSeg and CPM datasets. BoNuS [45] learns nuclei interior and boundary information through a boundary mining loss and detects missing nuclei using a curriculum-learning-based detection module.

However, the pseudo-labels generated from point annotations have compromised boundary accuracy, introducing noise that can negatively affect model training. Directly utilizing the generated probability map for subsequent supervision may amplify this noise. Furthermore, effectively separating adjacent nuclei with similar color and shape features poses a difficult task in nuclei segmentation (as shown in Figure 2). Some methods [45,46] seek to leverage spatial information from point annotations for instance separation through center-point prediction. However, as the direct use of point annotations for training is not feasible, a common approach is to generate Gaussian masks around point coordinates for training. Nevertheless, determining an optimal Gaussian kernel radius for center-point predictions becomes challenging due to variations in nuclei morphologies and sizes across different organs in training images. Additionally, it is hard to achieve stable predictions in single-branch center-point predictions, often resulting in false positive and false negative predictions, which is harmful to instance separation.

To address these challenges, this paper introduces a new framework that integrates three key components: the multi-scale Gaussian kernel module, the point-guided attention module, and the pseudo-label updating module. These components serve specific purposes: the multi-scale Gaussian kernel module distinguishes instances, the point-guided attention module reduces the impact of pseudo-label noise, and the pseudo-label updating module enhances the quality of pseudo-labels. Our contributions are summarized as follows:To tackle the difficulties in setting the Gaussian kernel radius and the instability of center-point predictions, we introduce a multi-scale Gaussian kernel mechanism, which utilizes multiple branches with different radii to accurately predict center points to separate adjacent nuclei.To mitigate noise from inaccurate pseudo-labels, we introduce a point-guided attention module. This enables the segmentation module to focus on learning features from the center-point prediction module, concentrating attention near true point labels.In addition, we introduce a module that optimizes pseudo-label boundaries using the exponential moving average (EMA) and k-means clustering, which leverages the model’s historical training information, point label information, and the color information of the training images to enhance the quality of pseudo-labels, aiming to improve segmentation performance.Our point-supervised approach has achieved state-of-the-art performance across various metrics in experiments conducted on three public datasets. Ablation studies have further validated the effectiveness of our modules.

## 2. Materials and Methods

In this section, the study is divided into three main modules, including the multi-scale Gaussian kernel module, the point-guided attention module, and the pseudo-label updating module. Figure 3 illustrates the framework proposed in this work. The entire model consists of the following modules:**Two types of decoders:** (1) the segmentation decoder branch for nuclei region detection; (2) the Gaussian decoder branch for center-point prediction. The labels used by the segmentation branch are generated from point labels via the Voronoi algorithm and k-means algorithm. Since the boundaries of these labels are predicted, they are referred to as pseudo-labels. The labels used by the Gaussian branches are generated by applying Gaussian expansion to each point in the point labels. All decoders share a common backbone.**Multi-scale Gaussian kernel module:** Multiple Gaussian branches are designed, each guided by a Gaussian mask with a different Gaussian expansion radius for training, with multi-branch predictions for center points. The center prediction map assists the segmentation branch in performing more accurate instance segmentation.**Pseudo-label updating module:** During training, the pseudo-labels used by the segmentation branch are iteratively optimized by this module in conjunction with historical training information to reduce the boundary noise caused by pseudo-labels.**Point-guided attention module:** The feature maps of each layer of the segmentation branch decoder pass through this module to learn features from the corresponding Gaussian branches. By focusing attention on areas near the real point labels, this module further reduces boundary noise.

Each module is described in detail in the subsequent sections.

### 2.1. Multi-Scale Gaussian Kernel Module

A multi-scale Gaussian kernel mechanism is designed to stabilize the prediction of center points. The multi-scale Gaussian kernel module comprises *k* Gaussian branches. For each branch, we utilize Gaussian masks with different radii as labels for center-point prediction. The formula for generating a Gaussian mask $M_{i}\left(x,y\right)$ is as follows:(1)$$M_{i}\left(x,y\right)\;\;=\;\;\left\{\begin{array}{cc} \exp\left(\frac{d_{i}^{2}\left(x,y\right)}{2\sigma^{2}}\right) & \mathrm{if}\;\;d\left(x,y\right)<r_{i},\\ 0 & \mathrm{otherwise}, \end{array}\right.$$
where σ denotes the Gaussian bandwidth, and $r_{i}$ represents the Gaussian kernel radius in the *i*-th Gaussian branch. $M_{i}\left(x,y\right)$ and $d_{i}\left(x,y\right)$ correspond to the Gaussian mask and the distance from point (x,y) to the nearest center point in the same branch, respectively.

The Gaussian branch is trained using weighted Mean Squared Error Loss. The specific loss function formula is as follows:(2)$$L_{gauss}=\frac{1}{k}\sum_{j=1}^{k}\frac{1}{\left|\Omega \right|}\sum_{i\in \Omega }w_{i}{\left(p_{i}^{j}-M_{i}^{j}\right)}^{2},$$
where Ω is the set of non-ignored pixels, and the variables $p_{i}^{j}$ and $M_{i}^{j}$ represent the pixel value at pixel *i* in the Gaussian prediction map and the Gauss label corresponding to the *j*-th Gaussian branch, respectively. $w_{i}$ is the weight of pixel *i*. Considering the imbalance between annotated points and background pixels, to enhance the influence factor of the foreground area and encourage more foreground predictions by the Gaussian branches, $w_{i}$ is set to 10 for pixels with a mask value greater than 0 and to 1 for background pixels. The final loss for the entire multi-scale Gaussian kernel module is obtained by averaging the loss of each branch.

The reason for choosing multi-radius and multi-branch approaches is that the sizes and morphologies of cells vary across different images, making it challenging to set a single appropriate Gaussian kernel radius. By employing multiple radii, the nuclei can adaptively find the most suitable radius for center-point prediction. Additionally, multi-branch predictions are more stable compared to single-branch predictions.

After obtaining the nuclear foreground probability map from the segmentation branch and the Gaussian prediction map from the Gaussian branch, a corresponding model inference strategy is designed. This strategy allows the predicted center points to fully guide and assist the segmentation network in performing more precise instance segmentation, effectively separating adjacent cells that are difficult to distinguish.

The specific strategy is illustrated in the testing phase part of Figure 3. From the Gaussian branch, we obtain *k* Gaussian prediction maps. Only pixels predicted as foreground by at least k/2 Gaussian branches are considered foreground. The final foreground map will contain several connected components. Noise regions with an area of fewer than 20 pixels are removed. For the remaining connected components, we compute their centroid to obtain the initial center-point prediction map *C*. For the generated segmentation prediction map, each connected component is assigned a unique instance label to generate the initial segmentation prediction instance map *S*.

Subsequently, the Voronoi algorithm is employed to generate the Voronoi map using the predicted center points. The Voronoi diagram aims to divide the plane into several polygonal regions based on a set of seed points. Each region contains one generating point, and all points within that region are closer to the generating point than to any other point in other regions.

Given a set of generating points in a plane, a Voronoi cell is defined as(3)$$V\left(p_{i}\right)=\left\{x\in R^{2}|\forall j\neq i,\;d\left(x,p_{i}\right)<d\left(x,p_{j}\right)\right\},$$
where $d(x,p_{i})$ denotes the Euclidean distance between point *x* and point $p_{i}$. Pixels in different regions of the Voronoi diagram *P* are marked with unique instance ids. Multiplying this map by the segmentation prediction instance map *S* produces the coarse instance map *I*. However, due to cell morphology variations and model prediction errors, not all nuclei are perfectly assigned to their respective Voronoi regions. To address this, we propose a model inference strategy for instance refinement, aiming to better align nuclei with their corresponding Voronoi regions.

Algorithm 1 outlines our model’s inference strategy. To ensure that each cell instance in the segmentation prediction map corresponds to a predicted centroid, we proceed as follows:For a given segmentation instance, if no centroids are found within it, we identify this as a false negative in the Gaussian prediction. Consequently, a centroid is added to the center of this instance. As shown in Figure 4a,b, assuming $S_{i}$ represents instances originally predicted correctly, they have been erroneously segmented into 4 parts due to the lack of corresponding centroids. This segmentation error is evident. However, after adding centroids, the Voronoi diagram generated from the new center-point map effectively segments it into the correct shape.In addition, Gaussian prediction centroids that fall within the background area of the segmentation prediction map are considered false positives and are discarded. The updated *C* is then used to generate *P* using the Voronoi algorithm.For each instance $S_{i}$ in *S*, the Voronoi map *P* divides the instance into several blocks. The regions occupying less than 10% of $S_{i}$’s total area typically represent errors caused by inaccurate Voronoi partition (e.g., $S_{i2},S_{i3},S_{i4},S_{i5}$ in Figure 4c). These are merged into the largest neighboring sub-region. Centroids not corresponding to any instance are considered false positives and are removed. The updated *C* is used to regenerate *P* and *I*, iterating until each centroid corresponds to one instance in *I*.
**Algorithm 1:** Inference Strategy
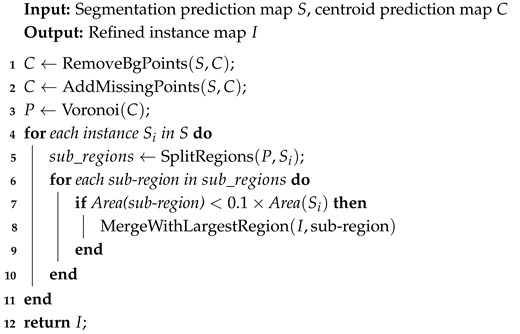


### 2.2. Pseudo-Label Updating Module

Since the model is trained using point annotations only, the Voronoi algorithm and k-means clustering are initially employed to generate the initial pseudo-labels for the segmentation branch. However, the quality of the labels’ boundaries generated by this method remains suboptimal. The pseudo-label updating module is illustrated in Figure 5. The entire pseudo-label updating strategy is divided into the training phase and the inference phase.

In the training phase, inspired by temporal ensembling [47] and Lin’s approach [44], we utilize the weight map to retain historical training information. Since the model has not yet learned sufficient features in the early stages of training, for the first 60 training epochs, we exclusively utilize the initially generated cluster labels for training (Voronoi labels are excluded from the label update process). The processed cluster label is a 3-value mask, where 0, 1, and 2 represent background, foreground, and ignored pixels, respectively. At the 60th epoch, the probability map *p* predicted by the model is used. This map is used to fill in the ignored pixels in the cluster labels K, generating the initial weight map $W_{0}$. The value of the pixel (x,y) in the map, $W_{0}\left(x,y\right)$, is generated as follows:(4)$$W_{0}\left(x,y\right)=\left\{\begin{array}{c} K\left(x,y\right)\;\;\;\;\mathrm{if}\;\;K\left(x,y\right)=0\;\;\mathrm{or}\;\;1,\\ p\left(x,y\right)\;\;\;\;\;\mathrm{if}\;\;K\left(x,y\right)=2. \end{array}\right.$$

During these steps, the exponential moving average (EMA) is incorporated for weight map updates. The EMA is chosen for pseudo-label updating because it gives more weight to recent predictions while smoothing out noise from earlier iterations. Compared to simple averaging or sliding window methods, the EMA better mitigates short-term fluctuations during updates, adapting progressively to dynamic changes in training. Additionally, its cumulative updating property enhances the stability and quality of pseudo-labels, which is crucial for improving model performance and reducing noise propagation in weakly supervised tasks. The formula can be expressed as follows:(5)$$W_{n}=\alpha p_{n}+\left(1-\alpha \right)W_{n-1},$$
where $p_{n}$ represents the probability map predicted by the model in the *n*-th epoch. The new weight map $W_{n}$ is generated by weighting the previous weight map $W_{n-1}$. After thresholding, the resulting weight map is employed to guide subsequent training. However, to prevent amplifying errors caused by prediction inaccuracies, pixels ignored by K continue to be ignored. The labels are updated at regular intervals.

After training, during the inference phase, image color information and point annotation details are integrated. For each pixel $x_{i}$, we combine the corresponding predicted probability map value $p_{i}$ with the RGB values $\left(r_{i},g_{i},b_{i}\right)$ of the image. Additionally, we include $d_{i}$, generated by the distance transform [45] from point annotations. These values together form the feature vector $f_{x_{i}}=\left({\hat{r}}_{i},{\hat{g}}_{i},{\hat{b}}_{i},{\hat{d}}_{i},{\hat{p}}_{i}\right)$, which is used for k-means clustering:(6)$$\underset{S}{arg\min}\sum_{i=1}^{k}\sum_{x_{i}\in S_{j}}{∥f_{x_{i}}-c_{j}∥}^{2}.$$

In this equation, *k* represents the number of clusters, $c_{j}$ denotes the center of the *j*-th cluster, and $S_{j}$ is the set of pixels assigned to the *j*-th cluster. The objective is to minimize the sum of squared distances between each pixel’s feature vector $f_{x_{i}}$ and the corresponding cluster center $c_{j}$. The 5 different variables are normalized or clipped to similar ranges to enhance clustering performance. The newly generated cluster labels will be used for the 2nd training cycle. The model produces its final results after the second iteration.

### 2.3. Point-Guided Attention Module

Although a pseudo-label update strategy was designed to improve the quality of pseudo-labels, boundary regions may still introduce noise, impacting model training. Therefore, a point-guided attention module is designed to focus the model on true point annotations, reducing the noise from pseudo-labels.

The specific framework is shown in Figure 6. The decoders for both the segmentation branch and the Gaussian branches are structurally identical, forming a U-shaped architecture [17] with the shared encoder. At each level, the segmentation branch feature map undergoes point-guided attention with the corresponding Gaussian feature maps. Subsequently, these feature maps are concatenated with the encoder’s feature map at the same level and upsampled to produce the feature map for the next level.

Similar to the approach in the Vision Transformer [19], the input feature maps are flattened into vectors for attention calculations. However, due to the large size of the feature maps, directly flattening them may result in high computational complexity. Therefore, the module splits the feature maps into 7×7 patches and performs point-guided attention on the corresponding patches from each branch before concatenating them. This patch-based approach enables parallel computation, thereby significantly enhancing the computational efficiency of the model. The mechanism for generating the Query, Key, and Value in the segmentation branch is as follows:(7)$$Q_{i}=L_{Q}\left(\frac{\sum_{j=1}^{k}g_{j}^{m_{i}}}{k}\right),K_{i}=L_{K}\left(\frac{\sum_{j=1}^{k}g_{j}^{m_{i}}}{k}\right),$$(8)$$V_{i}=L_{V}\left(S_{m_{i}}\right),$$(9)$$S_{m_{i}}=softmax\left(\frac{Q_{i}K_{i}^{T}}{\sqrt{d_{k}}}\right)V_{i},$$
where $Q_{i}$, $K_{i}$, and $V_{i}$ represent the Query, Key, and Value vectors for the *i*-th input, respectively, and *k* denotes the number of Gaussian branches. $d_{k}$ is the dimensionality of the Key vectors. $S_{m_{i}}$ indicates the i-th part of the segmentation feature, $g_{j}^{m_{i}}$ indicates the j-th Gaussian feature. $L_{Q}\left(\cdot \right)$, $L_{K}\left(\cdot \right)$, and $L_{V}\left(\cdot \right)$ denote linear projection for the input. To ensure the independence of predictions from each Gaussian branch, we exclusively perform self-attention calculations within each Gaussian branch, without sharing features with other branches.

### 2.4. Loss Function

The nuclei segmentation branch is trained jointly using both Voronoi labels and cluster labels. The corresponding losses, denoted by $L_{vor}$ and $L_{clu}$, both employ a combination of Dice Loss and Cross-Entropy (CE) Loss:(10)$$L_{vor},L_{clu}=L_{Dice}+L_{CE},$$(11)$$L_{Dice}=1-\frac{2\times \sum_{i=1}^{N}\left(p_{i}\times t_{i}\right)+\epsilon }{\sum_{i=1}^{N}p_{i}^{2}+\sum_{i=1}^{N}t_{i}^{2}+\epsilon },$$(12)$$L_{CE}=-\frac{1}{N}\sum_{i=1}^{N}\sum_{c=1}^{C}G_{i}^{C}\log S_{i}^{C},$$
where for $L_{\mathrm{dice}}$, $p_{i}$ represents the model’s prediction, $t_{i}$ is the ground-truth label, *N* is the number of pixels in the sample, and ε is a small regularization term to avoid division by zero. For $L_{\mathrm{CE}}$, *C* denotes the class corresponding to pixel *i*, and $G_{i}^{C}$ and $S_{i}^{C}$ represent the binary label and its corresponding predicted probability, respectively.

The overall loss is derived from the combined losses of the segmentation branch and the Gaussian branches:(13)$$L_{total}=\lambda_{1}L_{vor}+\left(2-\lambda_{1}\right)L_{clu}+\lambda_{2}L_{gauss},$$
where $\lambda_{1}$ and $\lambda_{2}$ represent loss weights.

## 3. Results

### 3.1. Datasets

Three datasets were employed to evaluate our method: the Monuseg dataset [48], the CPM17 dataset [49], and the CoNSeP dataset [25].

**Monuseg**: The Multi-Organ Nuclei Segmentation Challenge dataset is a public dataset downloaded from The Cancer Genome Atlas (TCGA), comprising histopathological images from nine organs. The dataset consists of 30 training images and 14 testing images, each with a resolution of 1000 × 1000 pixels, and a total of 28,846 nuclei.

**CPM17**: The Computational Precision Medicine dataset released in 2017 includes 64 H&E-stained images, each with the size of 500 × 500 or 600 × 600 pixels, containing a total of 7570 nuclei. The dataset is divided into 32 training and 32 testing images. The training and test sets both consist of images from four cancer types: glioblastoma multiforme (GBM), low-grade glioma (LGG), head and neck squamous cell carcinoma (HNSCC), and non-small-cell lung cancer (NSCLC), with eight images per cancer type.

**CoNSeP**: The Colorectal Nuclear Segmentation and Phenotypes dataset consists of 41 H&E-stained image tiles, each with a size of 1000 × 1000 pixels and a magnification of 40×. The images contain a total of 24,319 nuclei annotated by pathologists. The dataset is divided into 27 training images and 14 test images.

### 3.2. Evaluation Metrics

The model was evaluated using five metrics: object-level Dice coefficient ${\mathrm{Dice}}_{obj}$ [50], Aggregated Jaccard Index (AJI) [51], detection quality (DQ), segmentation quality (SQ), and panoptic quality (PQ) [25]. It should be mentioned that the ${\mathrm{Dice}}_{obj}$ measures the similarity between predicted and ground-truth objects in segmentation, focusing on the accurate detection of individual objects like nuclei or cells. These five evaluation metrics assess different aspects of model performance. Specifically, AJI and DQ mainly measure nuclei localization, while ${\mathrm{Dice}}_{obj}$ and SQ are focused on nuclei segmentation. PQ offers a comprehensive evaluation of the model’s overall detection quality.

### 3.3. Implementation Details

ResNet34 [52] was utilized as the backbone, and the Adam optimizer was employed for training, with a learning rate and weight decay set to 1 × 10^−4^. The batch size was set to 8, and the model was trained for 120 epochs. The same augmentation operations as described in Qu’s method [40] were performed. Each training image was divided into 16 overlapping patches of size 250 × 250 pixels. These patches were then transformed (cropped, rotated, etc.) to obtain image patches of size 224 × 224 pixels, which were used for model training. During testing, each image was cropped into 224 × 224 pixel patches with an overlap of 80 pixels. Empirically, we employed four Gaussian branches with radii set to 7, 9, 11, and 13. These parameters were chosen because when the Gaussian kernel radius is smaller than 7, most values in the Gaussian mask approach zero, making it difficult for the model to learn effective features. Conversely, when the radius exceeds 13, adjacent Gaussian kernels begin to overlap, leading to reduced accuracy in center-point prediction. The Gaussian bandwidth was defined as one-third of the corresponding Gaussian kernel radius. In the pseudo-label update strategy, the EMA weight α was set to 0.1. The weight map was updated every 10 epochs. In the loss function, the weights $\lambda_{1}$ and $\lambda_{2}$ were set to 1. The model was trained using an NVIDIA GeForce RTX 3090 GPU with PyTorch version 1.8.0 and CUDA version 11.2.

### 3.4. Comparative Results

Table 1 presents the comparative experimental results of our method against the most popular or highest-performing point-supervised and fully supervised methods, which we used as baselines. Our method surpasses the state-of-the-art (SOTA) by 3.4% in AJI and 5.1% in DQ metrics on Monuseg. On CPM17, it outperforms the SOTA by 2.4% in AJI and 5.0% in DQ. In the CoNSeP dataset, our method exceeds the SOTA by 2.8% in AJI and 0.9% in DQ. This indicates that our method effectively distinguishes instances. In terms of the ${\mathrm{Dice}}_{obj}$ and PQ metrics, our model surpasses the state-of-the-art by 2.4% and 3.2% on the MoNuSeg dataset, by 1.2% and 3.2% on the CPM17 dataset, and by 1.6% and 0.5% on the CoNSeP dataset, respectively.

However, due to the absence of nuclei boundary information in the labels used by our approach, there is still a gap between our method and fully supervised approaches in segmentation metrics such as ${\mathrm{Dice}}_{obj}$ and SQ. This is particularly evident in datasets with complex cell morphology and dense distributions, such as CoNSeP. The model focuses more attention on the areas near each nucleus’s center, which leads to a slight decrease in boundary precision and a minor drop in the SQ metric. However, this significantly improves the model’s ability to separate densely packed nuclei, which is essential for accurate cell counting, making this trade-off acceptable. Although there remains a gap between our model and fully supervised methods, our approach still outperforms most point-supervised methods across the majority of metrics.

The visual comparison results with some point-supervised methods are presented in Figure 7. In the yellow-highlighted regions, other methods fail to effectively differentiate closely adjacent cells, whereas our method demonstrates superior performance in distinguishing instances and boundaries, significantly improving the overall segmentation performance of the model.

We also evaluated the nuclei segmentation performance across different organs. As shown in Table 2, our model achieves high metrics for organs such as the bladder, brain, and lung. However, for the colon and breast, where the cells are typically elongated and densely packed, the performance is relatively poor and does not reach the average across the entire test set.

### 3.5. Ablation Studies

#### 3.5.1. Ablation Study on Three Modules

An ablation study is presented to validate the effectiveness of each module in our method. From Figure 8, it is evident that models B, D, E, and F, with the inclusion of the multi-scale Gaussian kernel module, exhibit the enhanced separation of adjacent cells. This improvement is particularly manifested in object-level metrics such as AJI and DQ in Table 3. The comparison between models A and C reveals that the introduction of the pseudo-label updating module results in improvements of 1.1%, 0.5%, and 4.0% in ${\mathrm{Dice}}_{obj}$ metrics on Monuseg, CPM17, and CoNSeP, respectively. There are also improvements in all other metrics. From the comparison between models B and D in Table 3, the introduction of point-guided attention leads to improvements of 2.3%, 1.0%, and 1.2% in AJI on Monuseg, CPM17, and CoNSeP, respectively. Finally, the combination of the three modules results in the optimal performance across most metrics.

#### 3.5.2. Ablation Study on the Loss Function

In the loss function, $\lambda_{1}$ and $\lambda_{2}$ are used to balance the weights between the segmentation branch and the Gaussian branch, respectively. For the segmentation branch, we directly adopted the loss function from [45]. An ablation study was conducted on these parameters to illustrate how their values influence segmentation performance. The best results are achieved when both $\lambda_{1}$ and $\lambda_{2}$ are set to 1.

The results of varying one parameter while keeping the other at its optimal value are shown in Figure 9. The first row illustrates the impact of $\lambda_{1}$ on performance when $\lambda_{2}$ is set to 1. When $\lambda_{1}$ is too small, the model performs poorly because the Voronoi labels contain center-point information, and using only cluster labels fails to separate close nuclei. Conversely, with a large $\lambda_{1}$, the model neglects boundary information from the cluster labels. The best performance occurs when $\lambda_{1}$ is around 1, so we use this value in subsequent experiments. The second row shows the effect of $\lambda_{2}$ when $\lambda_{1}$ is fixed at 1, with the best results in the range of 1 to 1.5. For simplicity, we use $\lambda_{2}=1$.

#### 3.5.3. Ablation Study on the Gaussian Branches

To investigate the impact of the multi-scale Gaussian kernel module on model prediction, the model was trained using 1 to 6 Gaussian branches separately. To prevent the Gaussian mask from being too small for training and the Gaussian kernels from merging when the radius is too large, integer values between 6 and 16 were selected for the Gaussian kernel radius in our experiments. As shown in Table 4, the experimental data for each branch were taken from the best results obtained with different Gaussian kernel radii. The metrics progressively improve as the number of Gaussian branches increases from 1 to around 4. However, as the number of branches continues to increase, the improvement is marginal. Consequently, we opt for using four Gaussian branches. It is worth noting that the kernel radius also affects the experiment, and more branches do not necessarily lead to better results. Nevertheless, a multi-branch and multi-scale design helps the model adapt to varying kernel parameters, reducing errors caused by inaccurate single predictions and enhancing overall robustness.

Figure 10 shows the impact of the number of Gaussian branches on model complexity, training time, and inference time. With each additional Gaussian branch, the model parameters and training time increase by approximately a factor of 1.1. In the inference phase, the inference time also grows by about 1.1 times. After the input images are processed by the model to produce outputs, additional time is required for instance partition of the output segmentation maps. During post-processing, we use a consolidated center prediction map from multiple Gaussian branches. The post-processing time is determined by the number of nuclei in the coarse instance predictions, rather than the number of Gaussian branches. This post-processing time accounts for approximately 70% of the total inference time.

#### 3.5.4. Ablation Study on Inference Strategy

The impact of the inference strategy on model performance was investigated across all three datasets. The results are presented in Table 5. It can be observed that, despite the longer post-processing time, the model achieved improvements of 0.7%, 0.1%, and 0.9% in ${\mathrm{Dice}}_{obj}$ scores on Monuseg, CPM17, and CoNSeP datasets, respectively. It should be noted that the inference strategy did not directly alter the foreground region predicted by the segmentation map, thus yielding no direct positive impact on the SQ metric.

Table 6 displays the inference time of different methods on the Monuseg dataset. Weak-Anno [40], WSPP [45], and SC-Net [44] directly use semantic segmentation maps to compute connected components for instance maps during inference; consequently, the inference time is short. However, these methods cannot effectively distinguish adjacent instances. SPN+IEN [43] performs better than previous methods; however, its post-processing operations are excessively time-consuming. Thanks to the Transformer’s superior parallel computing capabilities, our approach achieves an inference time of 0.513 s, comparable to existing point-supervised methods for generating a coarse instance map. Generating a refined instance map incurs a notable increase in time (2.044 s). However, our method maintains an inference time comparable to common instance segmentation networks like HoverNet (1.977 s) and Mask-RCNN (2.773 s). Additionally, it delivers improved performance across various metrics.

### 3.6. Impact of Pseudo-Label Updating Strategy

Figure 11 illustrates the impact of the pseudo-label updating strategy. Original clustering labels exhibit low accuracy in delineating cell nuclei boundaries, with significant areas of ignored pixels leading to wasted label information. While the updated pseudo-labels still exhibit discrepancies compared to the ground truth, their boundary precision has significantly improved relative to the initial pseudo-labels. In addition, the extent of ignored areas has been substantially reduced, positively influencing model training.

## 4. Discussion

In this study, we introduce a nuclei segmentation method based on point supervision, aiming to achieve a performance close to that of fully supervised instance segmentation with significantly lower annotation costs. Our method demonstrates a significant advantage in differentiating adjacent instances. In contrast to the semantic segmentation algorithms used for comparison [40,44,45], which rely solely on pseudo-labels, our approach utilizes Gaussian decoders to support the segmentation decoder through a point-guided attention mechanism and predicted center-point maps. This integration significantly improves the precision of differentiating adjacent instances. The effectiveness of this approach is corroborated by the results presented in Figure 7 and Table 3. Ablation experiments on the Gaussian decoder branches show that using multiple branches with varying radii for center-point prediction provides greater accuracy and stability compared to single-branch predictions. Moreover, the introduction of Gaussian decoders does not significantly increase the inference time, which remains comparable to that of common instance segmentation networks. This indicates that our method is practical for real-world clinical applications. Furthermore, as shown in Figure 11 and Table 3, our method effectively mitigates the impact of pseudo-label noise. The point-guided attention mechanism reduces dependency on pseudo-label boundary quality, while the pseudo-label updating strategy leverages historical training information to improve boundary accuracy. These enhancements not only enhance model performance but also accelerate convergence.

Our method also exhibits strong scalability and adaptability to other imaging modalities and datasets: (1) By requiring only simple point annotations for training, it significantly reduces annotation costs compared to fully supervised methods, making it inherently scalable and adaptable to other datasets or application scenarios. (2) The model does not rely on the characteristics of histopathological images. It is applicable to networks with dense instances and can be easily transferred to datasets from other imaging modalities, such as CT and ultrasound. (3) Due to the parallel computing advantages of the Transformer, the model can perform fast inference on large-scale datasets. This capability also enables the design of deeper and more complex network architectures, making it suitable for tackling more complex imaging tasks.

Our method can be integrated into clinical workflows in several ways. First, in clinical applications, our method reduces the annotation workload for doctors and specialists, allowing them to dedicate more time to diagnosis and treatment, which improves efficiency. Second, the method’s ability to work across different imaging modalities shows its potential for broader application in various medical environments. Third, the response time of our method is comparable to standard instance segmentation networks, offering timely feedback for medical diagnoses. Compared to other point-supervised segmentation algorithms, it provides more accurate nuclei localization to support diagnosis while requiring less training effort than fully supervised methods.

Although our method has shown promising results across multiple datasets, we acknowledge that there are still shortcomings in our approach. (1) First, our method demonstrates significant advantages in segmenting circular cells, as observed in test images of the stomach and prostate sections of the MonuSeg dataset. However, for elongated and densely packed cells, such as those in the liver and colon sections, the model performance is poorer. As shown in Figure 12, elongated cells are often predicted to be shorter than the ground truth, and overlapping regions in crowded areas are classified as background to ensure instance separation. The point-guided attention mechanism, together with the use of distance maps as feature vectors during pseudo-label clustering, enhances the model’s ability to differentiate instances. However, this approach may sacrifice some boundary accuracy. Therefore, improving the boundary quality of elongated cells will be a major focus of our future work. (2) Our aim is to maximize performance with weak labels. While computational cost reduction was not our primary goal, it remains a significant concern. We will further reduce model complexity and inference time to meet the requirements of practical clinical applications. (3) Although point annotations require significantly less effort compared to full annotations, marking the center point for every cell in larger datasets can still be a considerable burden. In future work, we will attempt to further reduce the amount of annotation needed, such as using partial points for incomplete annotation [45,54]. Additionally, we will design algorithms to minimize the model’s reliance on point annotations, alleviating the burden on pathologists and reducing application costs.

## 5. Conclusions

In this paper, we propose a nuclei segmentation algorithm based solely on point labels for annotation. Our approach uses a multi-scale Gaussian kernel mechanism and a point-guided attention module. These components effectively leverage spatial information from point labels, improving segmentation accuracy and reducing pseudo-label noise. Additionally, we introduce a pseudo-label updating strategy that integrates historical training data and image color information to enhance overall training performance. Our method achieves state-of-the-art performance among similar approaches on three public datasets. However, our method still has limitations, such as suboptimal performance in segmenting elongated cells, the need for further reduction in model complexity, and the potential to further reduce the amount of center-point annotation. We hope that our work, while maintaining good segmentation performance, can reduce the amount of data annotation to assist clinical practices.

## Figures and Tables

**Figure 1 bioengineering-12-00085-f001:**
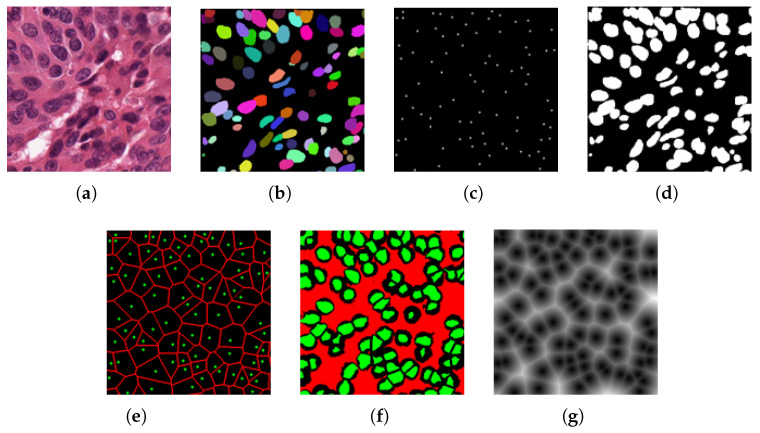
Different labels for the nuclei segmentation task. (**a**) Input image. (**b**) Pixel-level instance label. (**c**) Point annotation. (**d**) Binary mask. (**e**) Voronoi label. (**f**) Cluster label. (**g**) Distance map. In (**b**), different instances are marked with specific colors. In (**c**,**d**), white and black pixels, respectively, represent foreground and background regions, while in (**e**,**f**), green, red, and black pixels indicate foreground, background, and ignored areas, respectively. In (**g**), each pixel value represents the distance from that pixel to the nearest centroid, depicted as a grayscale image.

**Figure 2 bioengineering-12-00085-f002:**
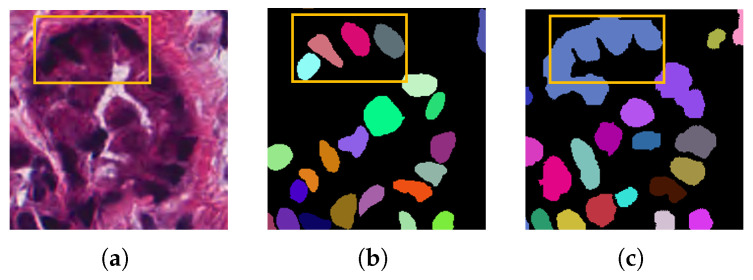
A bad case of segmenting adjacent nuclei. (**a**) Input image. (**b**) Ground truth. (**c**) Predicted instance map. As indicated within the yellow box, multiple closely adjacent nuclei are predicted as a single nucleus.

**Figure 3 bioengineering-12-00085-f003:**
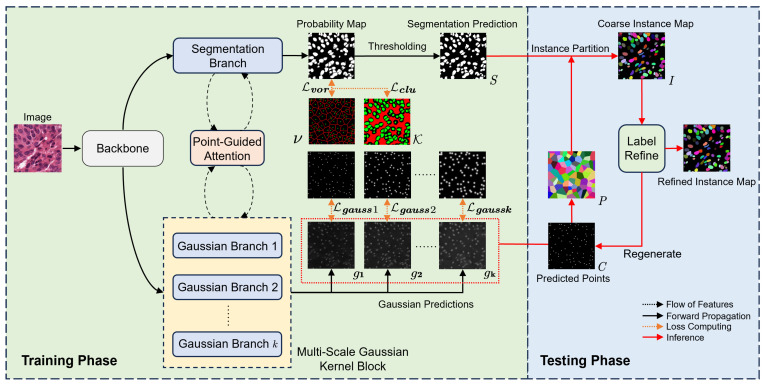
Overview of our proposed method. The Voronoi label ν, cluster label K, and Gaussian masks in the figure are all generated from point annotations. The model outputs two maps: (1) the center-point prediction from the Gaussian branch and (2) the segmentation prediction from the segmentation branch. During post-processing, the center-point map refines the segmentation map for more precise instance segmentation.

**Figure 4 bioengineering-12-00085-f004:**
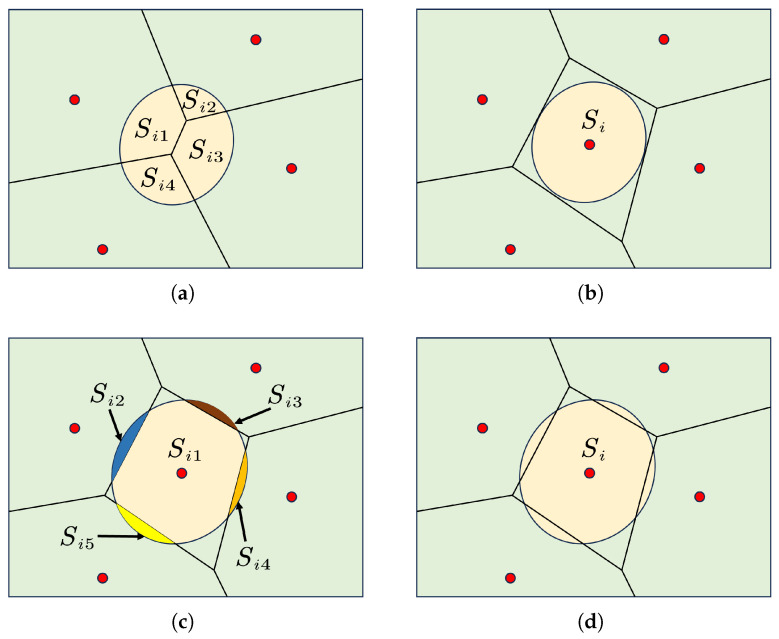
The issues arising in inference. The images in (**a**,**b**), respectively, depict the comparison result of segmentation before and after adding a centroid to an instance that does not correspond to a center point. The images in (**c**,**d**), respectively, illustrate the comparison result of nuclei segmentation before and after merging instances.

**Figure 5 bioengineering-12-00085-f005:**
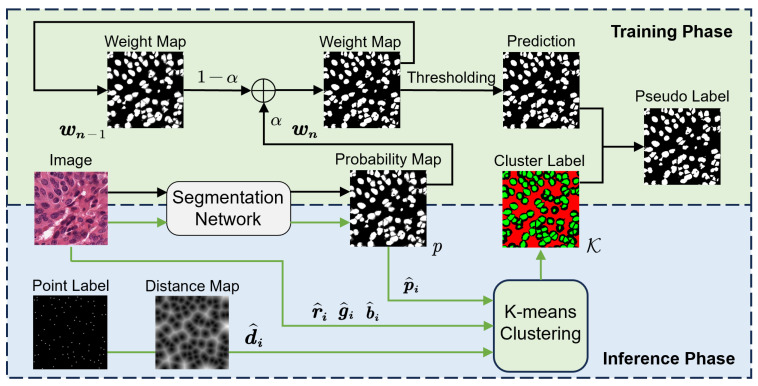
Pseudo-label update strategy. Black arrows denote label updates during a cycle of training, while green arrows represent updates after completing a training cycle.

**Figure 6 bioengineering-12-00085-f006:**
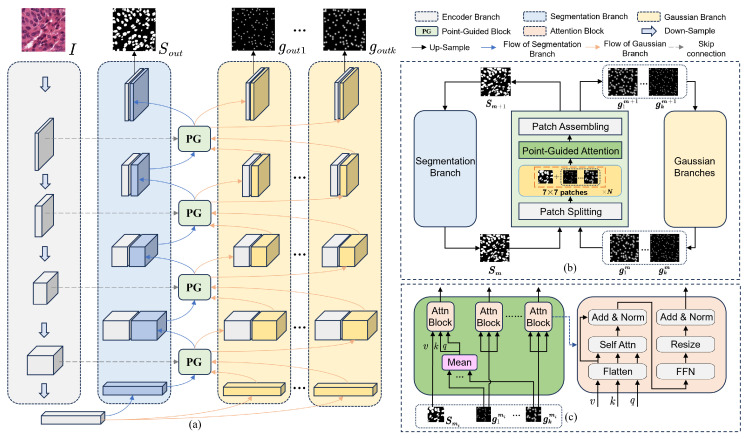
The structure of the point-guided attention module. (**a**) The interaction of the point-guided blocks within the model. (**b**) The processing of feature maps by the point-guided block in the segmentation branch and the Gaussian branches. (**c**) The main part of point-guided attention. Gray, blue, and yellow blocks indicate feature maps from the encoder, segmentation decoder, and Gaussian decoder, respectively. The abbreviations “Norm”, “Attn”, “PG”, and “FFN” refer to layer normalization, attention, point-guided block, and feed-forward network.

**Figure 7 bioengineering-12-00085-f007:**
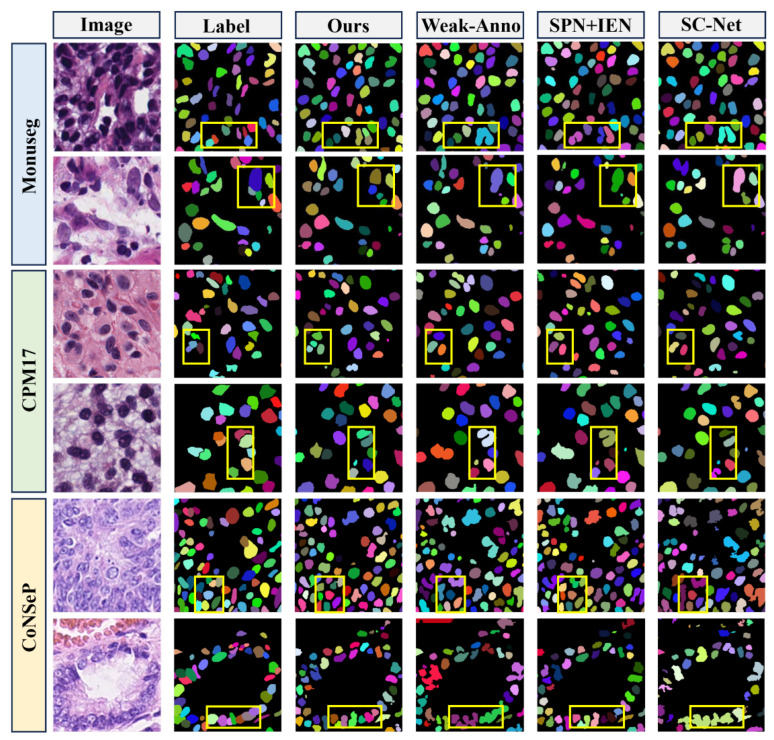
Segmentation results of different methods. The regions marked with yellow borders highlight the advantages of our approach. The test images are cropped from the Monuseg (first and second rows), CPM17 (third and fourth rows), and CoNSeP (fifth and sixth rows) datasets. Different nuclei are represented in different colors.

**Figure 8 bioengineering-12-00085-f008:**
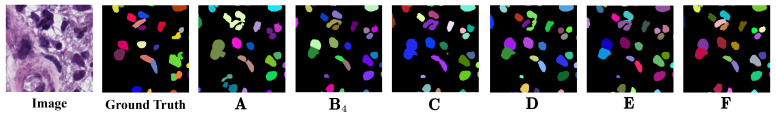
The results of ablation experiments. The letters below the figures correspond to the models listed in Table 3.

**Figure 9 bioengineering-12-00085-f009:**
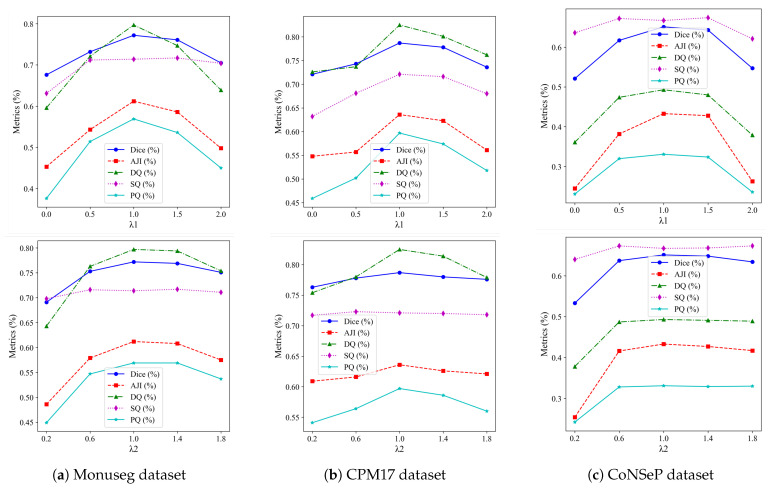
The results of using different $\lambda_{1}$ and $\lambda_{2}$ values across three datasets.

**Figure 10 bioengineering-12-00085-f010:**
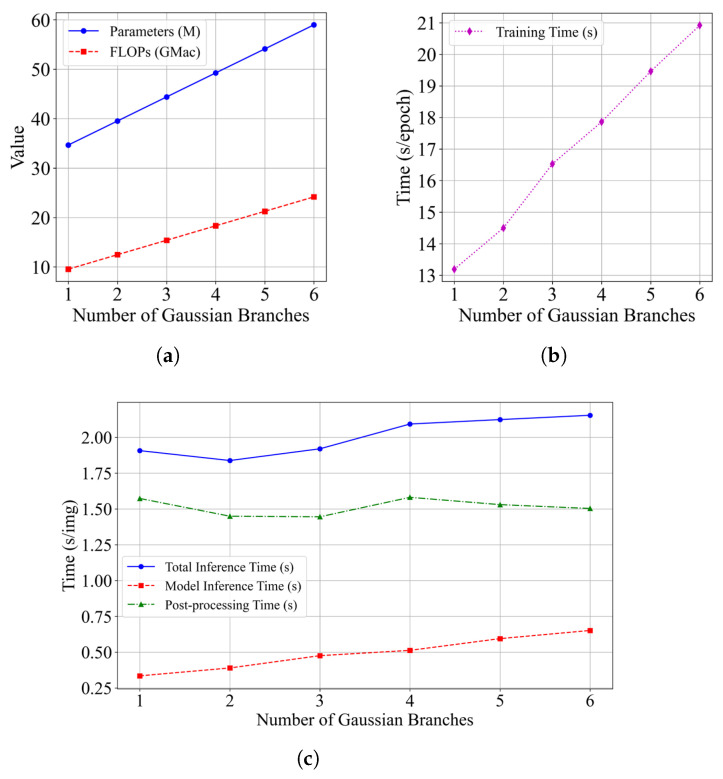
Quantitative results with different numbers of Gaussian branches on the Monuseg dataset. (**a**) Model complexity. (**b**) Training time. (**c**) Inference time.

**Figure 11 bioengineering-12-00085-f011:**
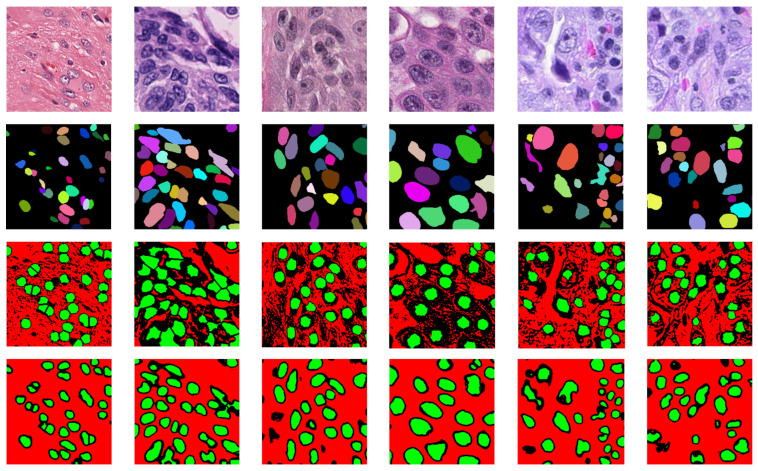
Visualization results of the pseudo-label updating strategy. Rows 2, 3, and 4 depict the ground truth, initial clustering labels, and updated clustering labels obtained after one round of training, where green, red, and black represent nuclei regions, non-nuclei regions, and uncertain regions, respectively.

**Figure 12 bioengineering-12-00085-f012:**
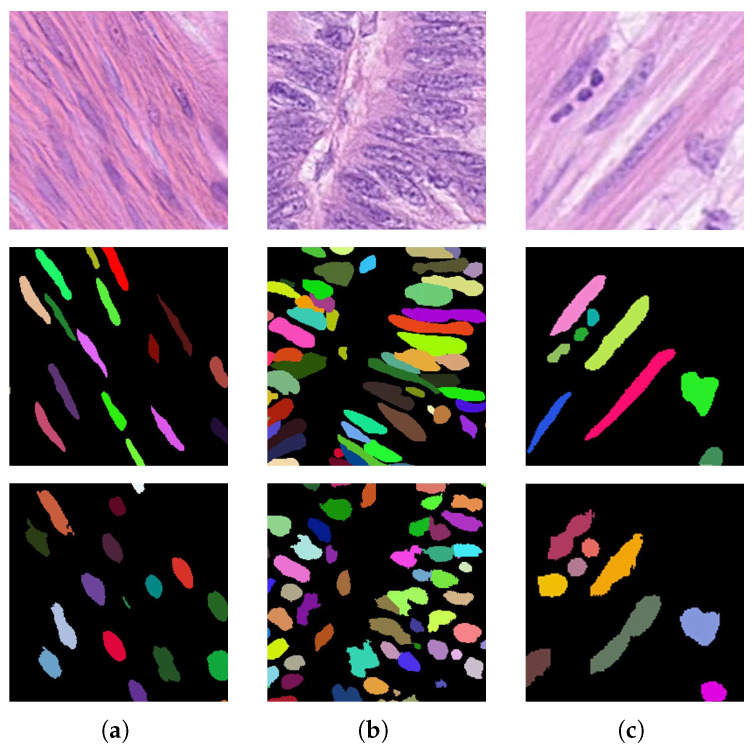
Some failure cases occur in our method when dealing with elongated and densely packed cells; a successful case is also shown. (**a**) Missed detection. (**b**) Failed segmentation of densely packed elongated cells. (**c**) A successful case.

**Table 1 bioengineering-12-00085-t001:** Comparative experiments on the Monuseg, CPM17, and CoNSeP datasets.

Methods	Monuseg					CPM17					CoNSeP				
	** ${\mathrm{Dice}}_{\mathrm{obj}}$ **	**AJI**	**DQ**	**SQ**	**PQ**	** ${\mathrm{Dice}}_{\mathrm{obj}}$ **	**AJI**	**DQ**	**SQ**	**PQ**	** ${\mathrm{Dice}}_{\mathrm{obj}}$ **	**AJI**	**DQ**	**SQ**	**PQ**
	**Fully Supervised**														
Mask-RCNN [53]	0.807	0.623	0.806	0.761	0.614	0.837	0.686	0.847	0.801	0.680	0.732	0.505	0.626	0.711	0.446
Micro-Net [18]	0.793	0.603	0.760	0.757	0.576	0.845	0.674	0.838	0.782	0.657	0.756	0.531	0.609	0.747	0.455
HoverNet [25]	0.817	0.618	0.770	0.773	0.597	0.848	0.705	0.854	0.814	0.697	0.807	0.571	0.702	0.778	0.547
	**Point-Supervised**														
Weak-Anno [40]	0.729	0.546	0.714	0.717	0.513	0.771	0.607	0.750	0.716	0.542	0.605	0.353	0.458	0.679	0.312
PseudoEdgeNet [42]	0.711	0.506	0.637	0.712	0.454	0.728	0.553	0.733	0.639	0.469	0.557	0.270	0.391	0.684	0.269
C2FNet [35]	0.717	0.539	0.701	0.715	0.501	0.735	0.567	0.657	0.683	0.448	0.569	0.259	0.429	0.683	0.293
WSPP [45]	0.733	0.546	0.724	0.697	0.506	0.746	0.561	0.742	0.689	0.517	0.609	0.358	0.460	0.683	0.315
SPN+IEN [43]	0.748	0.578	0.746	**0.718**	0.537	0.775	0.612	0.775	**0.726**	0.565	0.635	0.405	0.484	0.669	0.326
SC-Net [44]	0.738	0.562	0.730	0.712	0.521	0.759	0.598	0.746	0.706	0.530	0.609	0.372	0.436	**0.695**	0.305
**Ours**	**0.772**	**0.612**	**0.797**	0.714	**0.569**	**0.787**	**0.636**	**0.825**	0.721	**0.597**	**0.651**	**0.433**	**0.493**	0.667	**0.331**

**Table 2 bioengineering-12-00085-t002:** The segmentation results on the MonuSeg test set across different organs.

Organ	${\mathrm{Dice}}_{\mathrm{obj}}$	AJI	DQ	SQ	PQ
Bladder	0.787	0.635	0.819	0.721	0.590
Brain	0.781	0.620	0.786	0.707	0.556
Breast	0.754	0.591	0.762	0.696	0.530
Colon	0.731	0.544	0.719	0.698	0.502
Kidney	0.777	0.623	0.820	0.723	0.593
Lung	0.776	0.624	0.834	0.718	0.599
Prostate	0.774	0.610	0.785	0.726	0.570

**Table 3 bioengineering-12-00085-t003:** Ablation study on the proposed methods, where “MG”, “PA”, and “LU” represent the multi-scale Gaussian kernel module, point-guided attention module, and pseudo-label updating module. Results are presented as mean ± standard deviation from 5 runs.

	MG	PA	LU	Monuseg					CPM17					CoNSeP				
				** ${\mathrm{Dice}}_{\mathrm{obj}}$ **	**AJI**	**DQ**	**SQ**	**PQ**	** ${\mathrm{Dice}}_{\mathrm{obj}}$ **	**AJI**	**DQ**	**SQ**	**PQ**	** ${\mathrm{Dice}}_{\mathrm{obj}}$ **	**AJI**	**DQ**	**SQ**	**PQ**
A	**✕**	**✕**	**✕**	0.726 ± 0.012	0.534 ± 0.022	0.695 ± 0.021	0.714 ± 0.003	0.496 ± 0.018	0.764± 0.007	0.597 ± 0.010	0.742 ± 0.008	0.712 ± 0.004	0.531 ± 0.011	0.582 ± 0.023	0.337 ± 0.016	0.440 ± 0.018	0.672 ± 0.007	0.296 ± 0.016
B	**🗸**	**✕**	**✕**	0.742 ± 0.005	0.571 ± 0.003	0.729 ± 0.003	0.717 ± 0.002	0.523 ± 0.004	0.771 ± 0.003	0.612 ± 0.005	0.780 ± 0.003	0.716 ± 0.002	0.558 ± 0.003	0.629 ± 0.010	0.415 ± 0.005	0.467 ± 0.006	0.673 ± 0.003	0.316 ± 0.006
C	**✕**	**✕**	**🗸**	0.739 ± 0.007	0.559 ± 0.007	0.733 ± 0.006	0.719 ± 0.004	0.528 ± 0.007	0.772 ± 0.004	0.613 ± 0.002	0.799 ± 0.004	0.715 ± 0.003	0.573 ± 0.006	0.633 ± 0.012	0.353 ± 0.014	0.446 ± 0.13	0.675 ± 0.006	0.302 ± 0.011
D	**🗸**	**🗸**	**✕**	0.761 ± 0.003	0.593 ± 0.004	0.769 ± 0.005	0.717 ± 0.002	0.552 ± 0.006	0.776 ± 0.002	0.625 ± 0.002	0.804 ± 0.003	0.718 ± 0.002	0.579 ± 0.005	0.632 ± 0.006	0.425 ± 0.007	0.481 ± 0.005	0.666 ± 0.007	0.319 ± 0.002
E	**🗸**	**✕**	**🗸**	0.753 ± 0.004	0.581 ± 0.005	0.738 ± 0.006	**0.719 ± 0.003**	0.531 ± 0.006	0.780 ± 0.003	0.625 ± 0.001	0.807 ± 0.003	0.715 ± 0.002	0.579 ± 0.005	0.641 ± 0.006	0.423 ± 0.007	0.454 ± 0.006	**0.677 ± 0.004**	0.308 ± 0.006
F	**🗸**	**🗸**	**🗸**	**0.770 ± 0.002**	**0.609 ± 0.003**	**0.793 ± 0.004**	0.715 ± 0.003	**0.565 ± 0.004**	**0.785 ± 0.002**	**0.634 ± 0.002**	**0.821 ± 0.004**	**0.718 ± 0.003**	**0.592 ± 0.005**	**0.647 ± 0.004**	**0.428 ± 0.005**	**0.488 ± 0.005**	0.667 ± 0.004	**0.327 ± 0.004**

**Table 4 bioengineering-12-00085-t004:** Ablation study on the number of Gaussian branches.

Num	Monuseg					CPM17					CoNSeP				
	** ${\mathrm{Dice}}_{\mathrm{obj}}$ **	**AJI**	**DQ**	**SQ**	**PQ**	** ${\mathrm{Dice}}_{\mathrm{obj}}$ **	**AJI**	**DQ**	**SQ**	**PQ**	** ${\mathrm{Dice}}_{\mathrm{obj}}$ **	**AJI**	**DQ**	**SQ**	**PQ**
1	0.742	0.561	0.723	0.719	0.520	0.772	0.596	0.758	0.713	0.541	0.619	0.376	0.463	0.674	0.313
2	0.746	0.572	0.728	0.718	0.523	0.773	0.614	0.781	0.716	0.559	0.639	0.419	0.472	0.676	0.320
3	0.746	0.573	0.730	0.718	0.523	0.774	0.616	0.781	0.717	0.560	0.640	0.421	0.475	0.676	0.322
4	**0.747**	**0.574**	**0.732**	**0.719**	**0.527**	0.774	**0.617**	**0.783**	**0.718**	0.561	0.639	0.420	0.473	**0.676**	0.322
5	0.746	0.572	0.731	0.719	0.525	0.774	0.616	0.782	0.717	0.561	**0.641**	**0.422**	**0.476**	0.675	**0.323**
6	0.746	0.571	0.731	0.719	0.525	**0.775**	0.617	0.782	0.718	**0.562**	0.640	0.419	0.472	0.676	0.321

**Table 5 bioengineering-12-00085-t005:** Ablation study on the inference strategy. For each dataset, the first row represents the results using the coarse instance map obtained by directly multiplying the segmentation map *S* with the Voronoi partition *P* derived from the coarse center-point map *C*. The second row represents the results using the refined instance maps obtained from Algorithm 1.

Dataset	Method	${\mathrm{Dice}}_{obj}$	AJI	DQ	SQ	PQ
Monuseg	**✕**	0.765	0.592	0.790	**0.716**	0.566
	**🗸**	**0.772**	**0.612**	**0.797**	0.714	**0.569**
CPM17	**✕**	0.786	0.636	0.820	0.720	0.593
	**🗸**	**0.787**	**0.639**	**0.825**	**0.721**	**0.597**
CoNSeP	**✕**	0.642	0.402	0.488	**0.669**	0.329
	**🗸**	**0.651**	**0.433**	**0.493**	0.667	**0.331**

**Table 6 bioengineering-12-00085-t006:** The training and inference times on the Monuseg dataset. The parameter count in the table refers to the total number of parameters during the model training phase. The last two rows represent the times of our method in generating a coarse instance map and a refined instance map, respectively.

Methods	Params	Training	Total	Average
		**Time**	**Inference Time**	**Inference Time**
	**[M]**	**[s/epoch]**	**[s]**	**[s/img]**
HoverNet [25]	45.68	16.25	27.69	1.978
Micro-Net [18]	25.83	13.17	11.65	0.832
Mask-RCNN [53]	43.98	26.58	38.82	2.773
Weak-Anno [40]	**24.91**	**11.78**	**3.402**	**0.243**
WSPP [45]	49.82	12.06	5.348	0.382
SC-Net [44]	69.06	23.58	9.002	0.506
SPN+IEN [43]	49.86	43.28	84.42	6.030
Ours(C)	49.25	17.82	7.182	0.513
Ours(R)	49.25	17.82	28.62	2.044

## Data Availability

The datasets generated or analyzed during the study are available from the corresponding author upon reasonable request.

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
