# Peer review of "Weakly Supervised Nuclei Segmentation with Point-Guided Attention and Self-Supervised Pseudo-Labeling"

_bioengineering, 2025, doi:10.3390/bioengineering12010085_

Round 1
Reviewer 1 Report
Comments and Suggestions for Authors
The article introduces a weakly-supervised nuclei segmentation method utilizing point annotations and a self-supervised pseudo-labeling approach. It features a point-guided attention mechanism to reduce noise from pseudo-labels and enhances segmentation performance. The method achieves state-of-the-art results on three public datasets, demonstrating its effectiveness in reducing the need for manual annotations in histopathological image analysis.
Overall, the paper is a valuable contribution to the field of medical image analysis.
Comments on the Quality of English Language1. Why does it describe as "nuclei segmentation" or "nucleus segmentation" ? should they be maintained consistency in terminology?
2. The paper lacks experiments or explanations regarding the selection of parameters, such as why the loss function parameters λ1, λ2, and λ3 are simply set to 1 directly.
3. The authors do not explain why the SQ metric for Monuseg and CoNSeP shown in Table 3 did not improve, and even decreased, after applying the method described in the paper.
4. In the "Datasets" section, the introduction of Monuseg uses "1000×1000 pixels," but the introduction of CoNSeP uses "f 1,000×1,000 pixels." Please ensure consistency in the use of the thousand separator.
5. In line 53, the sentence 'However, these methods trade-off label accuracy.' uses 'trade-off' as a noun, which is not correct because nouns cannot function as verbs in this context. The sentence is incorrect. It could possibly be revised to 'However, these methods trade off label accuracy.' or 'However, these methods have a trade-off with label accuracy.'
6. In the paper, sentences like the one on line 86, 'To address these challenges, this paper introduces a framework that combines the multi-scale Gaussian kernel module, point-guided attention module, and pseudo-label updating module, which are employed to distinguish instances, mitigate the impact of pseudo-label noise, and enhance the quality of pseudo-labels, respectively,' are too long. The author should consider breaking them up for better readability.
Reviewer 2 Report
Comments and Suggestions for Authors
Good research paper
Author Response
We sincerely appreciate your positive evaluation of our manuscript. We hope that the revisions we have made based on the other reviewers' comments further enhance the clarity and impact of the paper. Thank you once again for your time and consideration.

Reviewer 3 Report
Comments and Suggestions for Authors
In this article, the author proposed a weakly supervised nuclei segmentation method with point-guided attention and self-supervised pseudo-labeling. Although the author tried to show the advantages of the proposed method through a large number of block diagrams and experiments, there are still some problems that need to be further solved, as follows:
1. The introduction lacks a review of weakly supervised deep networks and a survey of existing point-guided attention mechanisms.
2. The logical relationship in the aspect description is unclear. It is recommended that the author provide an overview of the method.
3. All formula characters used need to be clearly defined.
4. It is recommended that the authors discuss in detail the value of the balance parameter in the loss function.
5. The training and inference times of all segmentation methods should appear in the manuscript.
6. It is recommended that the authors include the limitations of the proposed method in the conclusion section.
7. The problem that the author is concerned about seems to be more reasonably solved by using a classification approach, which allows scene-level image interpretation to be achieved through a small number of labeled samples. For the idea of classification, see DOI: 10.1109/JSTARS.2021.3111740.
Comments on the Quality of English LanguageThe English could be improved to more clearly express the research.
Reviewer 4 Report
Comments and Suggestions for Authors
The manuscript explores a novel method for nuclei segmentation based on weak supervision using point-guided attention and pseudo-labeling. The topic is significant as it addresses the challenges of accurate segmentation in medical imaging with reduced annotation costs. The authors demonstrate state-of-the-art results across multiple datasets, which strengthens the manuscript. However, there are areas where clarification, elaboration, and revisions are necessary to enhance its clarity and scientific contribution.
Lines 1–20: The abstract effectively summarizes the key contributions and results. However, it would benefit from a more concise statement regarding the significance of the method in clinical applications.
Lines 21–30: The advantages of weakly-supervised methods over fully-supervised approaches are highlighted but could be expanded with specific examples or citations for added context.
Lines 31–50: The discussion of traditional segmentation methods would benefit from more recent references to reflect current trends in the field.
Line 122: Clarify how the Gaussian kernel radii were chosen for the multi-scale mechanism. Are these values dataset-specific, or can they generalize?
Lines 186–213: The pseudo-label updating mechanism is well-described, but the rationale for using exponential moving average (EMA) over other smoothing techniques needs elaboration.
Algorithm 1 (Lines 167–184): The inference strategy should include computational cost comparisons to justify its real-world applicability.
Lines 294–307: Comparative results with fully-supervised methods are promising. However, the authors should discuss potential limitations when scaling this method to larger datasets or more complex histopathological images.
Lines 381–400: While the discussion on boundary refinement is insightful, including visual examples of errors (e.g., elongated cells) alongside successful cases would provide better context.
Figures 3, 7, and 10: These figures illustrate key results but would benefit from clearer legends and more detailed captions, especially for non-expert readers.
Table 3 (Line 331): Provide statistical significance analysis to strengthen claims of improvement.
Line 106: Don’t use active sentences from the point of English structure.
Include a dedicated section or paragraph addressing the scalability and adaptability of the method to other imaging modalities or datasets.
Discuss potential integration of this method into clinical workflows and its advantages over existing segmentation tools.
Simplify technical jargon and include a glossary for terms like "EMA" and "Diceobj" for non-specialist readers.
Round 2
Reviewer 1 Report
Comments and Suggestions for Authors
The article introduces a weakly-supervised nuclei segmentation method utilizing point annotations and a self-supervised pseudo-labeling approach. It features a point-guided attention mechanism to reduce noise from pseudo-labels and enhances segmentation performance. The method achieves state-of-the-art results on three public datasets, demonstrating its effectiveness in reducing the need for manual annotations in histopathological image analysis.
Overall, the paper is a valuable contribution to the field of medical image analysis. Recommend for acceptance.
Reviewer 3 Report
Comments and Suggestions for Authors
No Comments.
Reviewer 4 Report
Comments and Suggestions for Authors
It can be publisable as it is.